# MiR-501-3p Forms a Feedback Loop with FOS, MDFI, and MyoD to Regulate C2C12 Myogenesis

**DOI:** 10.3390/cells8060573

**Published:** 2019-06-11

**Authors:** Lianjie Hou, Linhui Zhu, Huaqin Li, Fangyi Jiang, Lingbo Cao, Ching Yuan Hu, Chong Wang

**Affiliations:** 1Guangdong Provincial Key Laboratory of Agro-Animal Genomics and Molecular Breeding, National Engineering Research Center for Breeding Swine Industry, College of Animal Science, South China Agricultural University, Guangzhou 510642, Guangdong, China; scaujackhou@163.com (L.H.); zlhmieh@163.com (L.Z.); huaqinli007@stu.scau.edu.cn (H.L.); FangyiJiangSCAU@163.com (F.J.); caolingbo2014@163.com (L.C.); 2Department of Human Nutrition, Food and Animal Sciences, College of Tropical Agriculture and Human Resources, University of Hawaii at Manoa, Honolulu, HI 96822, USA; chinghu@hawaii.edu

**Keywords:** proliferation, myogenic differentiation, skeletal muscle, bioinformatics analysis

## Abstract

Skeletal muscle plays an essential role in maintaining body energy homeostasis and body flexibility. Loss of muscle mass leads to slower wound healing and recovery from illness, physical disability, poor quality of life, and higher health care costs. So, it is critical for us to understand the mechanism of skeletal muscle myogenic differentiation for maintaining optimal health throughout life. miR-501-3p is a novel muscle-specific miRNA, and its regulation mechanism on myoblast myogenic differentiation is still not clear. We demonstrated that FOS was a direct target gene of miR-501-3p, and MyoD regulated miR-501-3p host gene Clcn5 through bioinformatics prediction. Our previous laboratory experiment found that MDFI overexpression promoted C2C12 myogenic differentiation and MyoD expression. The database also showed there is an FOS binding site in the MDFI promoter region. Therefore, we hypothesize that miR-501-3p formed a feedback loop with FOS, MDFI, and MyoD to regulate myoblast differentiation. To validate our hypothesis, we demonstrated miR-501-3p function in the proliferation and differentiation period of C2C12 cells by transfecting cells with miR-501-3p mimic and inhibitor. Then, we confirmed there is a direct regulatory relationship between miR-501-3p and FOS, MyoD and miR-501-3p, FOS and MDFI through QPCR, dual-luciferase reporter system, and ChIP experiments. Our results not only expand our understanding of the muscle myogenic development mechanism in which miRNA and genes participate in controlling skeletal muscle development, but also provide treatment strategies for skeletal muscle or metabolic-related diseases in the future.

## 1. Introduction

Skeletal muscle comprises about 45% of the human body mass. Aging and disease also accompany skeletal muscle loss (muscle atrophy) [1]. Loss of muscle mass results in slower wound healing and recovery from illness, physical disability, more inferior quality of life, and higher health care costs [2]. Skeletal muscle is not only crucial for physical performance but is also a significant contributing factor for maintaining energy homeostasis [2,3]. Skeletal muscle is involved in many metabolic pathways, such as the insulin-stimulated glucose uptake, fatty acid metabolism, and glycogen synthesis [4]. Skeletal muscle metabolic syndrome would lead to insulin resistance and obesity. Since skeletal muscle plays an essential role in maintaining body flexibility and energy homeostasis, it is critical for us to understand the mechanism of skeletal muscle myogenic differentiation for maintaining optimal health throughout life.

miRNA is a small noncoding RNA molecule found in plants, animals, and some viruses [5]. miRNA exerts its function in mRNA silencing by binding to the complementary sequences within its target RNA [6,7]. For example, mice skeletal muscle-specific miR-1 and miR-133 are clustered on the same gene loci and transcribed together in a tissue-specific manner, but miR-1 promotes myogenesis through repressing histone deacetylase 4, whereas miR-133 enhances proliferation by inhibiting serum response factor [8]. miR-501-3p is a novel muscle-specific miRNA and expresses specifically in activated myogenic progenitor cells and newly formed myofibers, and miR-501 level in mdx mice serum correlates to the regenerative phase of muscle [9]. However, this study did not examine the miR-501 regulation mechanism on myoblast proliferation and differentiation during myogenic differentiation. 

Through bioinformatics prediction, we discovered that FOS was a direct target gene of miR-501-3p. FOS, a proto-oncogene, forms the heterodimer complex Activator Protein-1 (AP-1) with C-JUN [10]. AP1 binds to the specific DNA sequence at the promoter and enhancer of target genes, then converts extracellular signals into cellular gene expression [11]. FOS inhibits mice myoblast myogenesis by binding to the putative cAMP response element on the MyoD promoter and inhibiting MyoD expression in dividing myoblasts [12]. Conversely, MyoD binding site overlaps with the serum-responsive element in the FOS promoter, then MyoD acts as a negative regulator for FOS transcription by blocking serum responsiveness through this binding site [13]. These studies indicate the existence of a feedback loop between FOS and MyoD to regulate myoblast differentiation. In addition, through analysis of MyoD ChIP-seq data, we found MyoD regulates miR-501-3p host gene transcription. Previous experiments from our laboratory found that MDFI overexpression promoted C2C12 myoblast myogenic differentiation and MyoD expression. The database shows there is an FOS binding site in the MDFI promoter region. Therefore, we hypothesize that miR-501-3p and MDFI are also involved in the feedback regulation loop between MyoD and FOS to regulate myoblast differentiation. To validate our hypothesis, we demonstrated miR-501-3p function in the proliferation and differentiation period of C2C12 cells by transfecting cells with miR-501-3p mimic and inhibitor. Then, we confirmed there is a direct regulatory relationship between MyoD and miR-501-3p, FOS and MDFI through QPCR, dual-luciferase reporter system, and ChIP experiments. These results expand our understanding of how miRNAs and genes work collaboratively in regulating skeletal muscle development.

## 2. Results

### 2.1. The Dynamic Changes of miR-501-3p Level during C2C12 Myogenic Differentiation

Before elucidating the function and regulatory mechanism of miR-501-3p in C2C12 myogenic differentiation, we measured the relative miR-501-3p level at different stages of C2C12 myogenic development. The QRT-PCR result shows miR-501-3p level was upregulated (*p* < 0.05) during the C2C12 cell proliferation, but the miR-501-3p level was downregulated (*p* < 0.05) during the C2C12 differentiation (Figure 1A). The QRT-PCR result shows on day 1 of the C2C12 proliferation (P1, Figure 1B) and day 7 of differentiation (D7, Figure 1C) that miR-501-3p was successfully overexpressed (*p* < 0.001) or inhibited (*p* < 0.001) after transfection with either miR-501-3p mimics or inhibitor, respectively.

### 2.2. MiR-501-3p Inhibits C2C12 Proliferation

MiR-501-3p mimics and inhibitor were transfected into C2C12 for 24 h. MiR-501-3p mimics reduced (*p* < 0.001) the percentage of Edu-positive cells and miR-501-3p inhibitor increased (*p* < 0.001) the percentage of Edu-positive cells (Figure 2A,B). In addition, we analyzed cells at different phases by flow cytometer. The results show that miR-501-3p mimics caused cell arrest in the G2 phase (*p* < 0.01), while miR-501-3p inhibitor increased (*p* < 0.01) the percentage of cells in the S phase (Figure 2C). The QRT-PCR result shows that miR-501-3p mimics increased (*p* < 0.001) CCNB mRNA level, and had no effect on the mRNA level of CCNA, CDK1, and CDK2 (Figure 2D). mRNA level of CCNA was increased (*p* < 0.001) and there was no change for the mRNA level of CCNB, CDK1, and CDK2 after transfection with miR-501-3p inhibitor (Figure 2D). Western blot results confirm that miR-501-3p mimics increased (*p* < 0.05) CCNB protein level, and miR-501-3p inhibitor increased (*p* < 0.05) CCNA protein level (Figure 2E). The relative protein levels obtained from WB bands gray scanning are presented in Appendix A. These results show miR-501-3p inhibited C2C12 proliferation.

### 2.3. MiR-501-3p Promotes C2C12 Differentiation

C2C12 were transfected with miR-501-3p mimics or inhibitor and cultured in the differentiation medium for seven days. We confirmed the miR-501-3p effect on C2C12 differentiation by the anti-Myosin immunofluorescent assay. Overexpressing miR-501-3p increased (*p* < 0.01) the Myosin-positive cells while inhibiting miR-501-3p lowered (*p* < 0.001) the Myosin-positive cells (Figure 3A,B). Overexpression of miR-501-3p elevated the mRNA level of MyoG (*p* < 0.001) and Myosin (*p* < 0.001, Figure 3C), and miR-501-3p inhibitor decreased the mRNA level of MyoG (*p* < 0.001) and Myosin (*p* < 0.001, Figure 3D). Western blot result shows overexpression of miR-501-3p also elevated protein level of MyoG (*p* < 0.05) and Myosin (*p* < 0.001), and miR-501-3p inhibitor decreased the protein level of MyoG (*p* < 0.05) and Myosin (*p* < 0.01, Figure 3E). The relative protein levels obtained through WB bands gray scanning are presented in Appendix A. These results show that miR-501-3p promotes C2C12 differentiation.

### 2.4. FOS Is a Direct Target Gene of miR-501-3p

To further investigate the potential mechanism by which miR-501-3p regulates the C2C12 myogenesis, we need to identify the target gene of miR-501-3p. Using bioinformatics analysis and the sequence alignment, we found that the miR-501-3p seed sequence is highly conserved (Figure 4A), and the miR-501-3p complementary seed sequence in the FOS 3′UTR is also highly conserved (Figure 4B). So there may be a potential binding site in FOS 3′UTR for miR-501-3p (Figure 4C). To verify the targeted relationship between miR-501-3p and FOS, we first measured the mRNA level of FOS using qRT-PCR in the C2C12 transfected with miR-501-3p mimics or inhibitor. The results show that overexpression of miR-501-3p decreased (*p* < 0.001) FOS mRNA level while inhibiting miR-501-3p increased (*p* < 0.001) the FOS mRNA level (Figure 4D). The WB results confirmed the qRT-PCR results (Figure 4E). The relative protein levels obtained through WB bands gray scanning are presented in Appendix A. Subsequently, we constructed the pmirGLO-*FOS* 3′UTR recombinant vector containing the binding site of miR-501-3p. We found that relative luciferase activity was decreased (*p* < 0.01; Figure 4F) when HEK-293T cells were co-transfected with miR-501-3p mimics and pmirGLO-*FOS*-3′UTR. However, miR-501-3p mimics did not affect mutated pmirGLO-*FOS*-3′UTR relative luciferase activity (Figure 4F). This result indicates that FOS is the direct target gene of miR-501-3p.

### 2.5. Overexpressing FOS Promotes C2C12 Proliferation

Since FOS is the direct target gene of miR-501-3p, we explored the role of FOS in C2C12 myogenic development. We overexpressed and inhibited FOS expression through transfecting C2C12 with pcDNA3.1(+)-FOS recombinant vector and siRNA-FOS, respectively (Figure 5A). Overexpressing FOS increased (*p* < 0.001) the percentage of Edu-positive cells and siRNA-FOS decreased (*p* < 0.001) the percentage of Edu-positive cells (Figure 5B,C). The qRT-PCR result shows pcDNA3.1(+)-FOS recombinant vector increased (*p* < 0.001) the mRNA level of CCNA and had no effect on the mRNA level of CCNB, CDK1, and CDK2 (Figure 5D). After C2C12 were transfected with siRNA-FOS, we found siRNA-FOS decreased (*p* < 0.05) the mRNA level of CCNB mRNA level, and did not affect the mRNA level of CCNA, CDK1, and CDK2 (Figure 5D). Western blot result confirmed that pcDNA3.1(+)-FOS recombinant vector increased protein level of CCNA (*p* < 0.05, Figure 4D), and siRNA-FOS increased CCNB (*p* < 0.01) protein level (Figure 4D). The relative protein levels obtained through WB bands gray scanning are presented in Appendix A. From the FOS overexpression and inhibition results, we conclude that FOS promotes C2C12 proliferation.

### 2.6. Overexpressing FOS Inhibits C2C12 Differentiation

We then determined the role of FOS in C2C12 differentiation. After seven days differentiation, the C2C12 FOS mRNA level was still upregulated (*p* < 0.001) or repressed (*p* < 0.001) by transfecting with either pcDNA3.1(+)-FOS recombinant vector or siRNA-FOS, respectively (Figure 6A). We used the anti-Myosin immunofluorescent assay to measure the differentiation of C2C12. Overexpressing FOS decreased (*p* < 0.001) the Myosin-positive cells while inhibiting FOS increased (*p* < 0.001) the Myosin-positive cells (Figure 6B,C). Overexpression of FOS lowered the mRNA level of MyoG (*p* < 0.001), and Myosin (*p* < 0.001, Figure 6D). Inhibiting FOS increased the mRNA level of MyoG (*p* < 0.001) and Myosin (*p* < 0.001, Figure 6D). Western blot result shows overexpression of FOS lowered the protein level of MyoG (*p* < 0.05) and Myosin (*p* < 0.01, Figure 6E). Inhibiting FOS increased the protein level of MyoG (*p* < 0.01) and Myosin (*p* < 0.01, Figure 6E). The relative protein levels obtained through WB bands gray scanning are presented in Appendix A. These results show FOS inhibits C2C12 differentiation.

### 2.7. MiR-501-3p Promotes C2C12 Myogenesis by Targeting FOS

Since miR-501-3p inhibits C2C12 proliferation and promotes C2C12 differentiation, and FOS is a direct target gene of miR-501-3p, we co-overexpressed miR-501-3p and FOS to verify whether FOS attenuated the miR-501-3p effect on C2C12 myogenic development. Through the Edu immunofluorescent staining, we found FOS weakens (*p* < 0.01) the miR-501-3p inhibition effect on C2C12 proliferation. The Myosin immunofluorescent staining results show FOS reduced (*p* < 0.01) the miR-501-3p promotion effect on C2C12 differentiation (Figure 7A,B), so we conclude that miR-501-3p regulates C2C12 myogenesis by targeting FOS.

### 2.8. MiR-501-3p Formed a Feedback Loop with FOS, MDFI, and MyoD 

Previous studies indicate the existence of a feedback loop between FOS and MyoD to regulate myoblast myogenic development. The MyoD ChIP-seq data shows that MyoD may regulate miR-501-3p host gene Clcn5 transcription (Figure 8A). Our laboratory also found that MDFI overexpression promoted C2C12 myoblast myogenic differentiation and MyoD expression [14]. The database shows there is an FOS binding site in the MDFI promoter region (Figure 8A). Therefore, we verified whether miR-501-3p and MDFI are also involved in the feedback regulation loop between MyoD and FOS to regulate C2C12 myogenic development. Through the ChIP results, we found FOS bound with a 120 bp long fragment amplified from the -496 upstream of MDFI transcription initiation site (Figure 8B). Then, we constructed the pGL3-basic-MDFI promoter recombinant vector (pGL3-basic-MDFI), and through the dual-luciferase reporter assay, we found FOS decreased the pGL3-basic-MDFI vector relative luciferase activity, but this inhibition was abolished by the mutated FOS binding site (TGACTCA) (Figure 8C). The ChIP results also show MyoD bound with a 100 bp long fragment amplified from the -40 upstream of Clcn5 transcription initiation site (Figure 8B). We also constructed the pGL3-basic-Clcn5 promoter recombinant vector (pGL3-basic-Clcn5), and through the dual-luciferase reporter assay, we found MyoD increased the pGL3-basic-Clcn5 vector relative luciferase activity, but this promotion was abolished by the mutated MyoD binding site (CAGCTGCTGC) (Figure 8D). Then, we overexpressed FOS or MyoD in C2C12, and through QPCR detection, we found FOS inhibited the expression of Mdfi and MyoD increased the expression of Clcn5 and miR-501-3p (Figure 8E). Finally, there is a feedback loop formed by miR-501-3p, FOS, MDFI, and MyoD to regulate myogenic differentiation (Figure 8F).

## 3. Discussion

Skeletal muscle represents about 40% of body weight. Skeletal muscle has the ability to contract and relax to produce skeletal movement. In addition, skeletal muscles play a critical role in body temperature homeostasis, protein reserve storage, and soft tissues protection [15]. Mammal embryonic skeletal muscle myogenesis consists of several major stages: dermomyotome and myotome myogenic progenitor cell proliferation and differentiation into myoblasts first, then myoblast differentiation and fusion into myotubes, and finally myotube differentiation into myofibers [16]. Skeletal muscle development is a complicated process which involves proper miRNAs and gene expression for myoblast proliferation, differentiation, migration, and death. These processes are predominantly regulated by MRFs (myogenic regulatory factors) family [17,18]. MRFs include MyoD, MYF5, myogenin, and MYF6. MyoD and MYF5 participate in the first stage of skeletal muscle development by promoting myogenic progenitor cell proliferation and differentiation into myoblasts. Myogenin plays a critical role in the myoblast differentiation into myotubes, and MYF6 participates in myoblast differentiation and cell fate determination [19,20].

MyoD is expressed at extremely low and undetectable levels in quiescent satellite cells, but the expression of MyoD is increased in activated satellite cells and myoblasts [21]. The increased MyoD expression removes cells from the cell cycle and promotes terminal differentiation by enhancing the transcription of p21 and myogenin [22]. Our results show MyoD bound to the promoter region of the miR-501-3p host gene Clcn5, and MyoD increased the transcription of Clcn5 and miR-501-3p, indicating MyoD exerts the cell cycle regulation function also through inhibiting the miR-501-3p target gene FOS expression.

FOS is an inherent gene in mammals and belongs to the immediate early response genes. The FOS gene is highly conserved, and its family also contains FOS1 and FOS2 [23]. FOS is rapidly expressed under the induction of various factors. As a nuclear protein transcription factor, FOS plays an essential role in regulating cell growth, division, proliferation, differentiation, and even programmed death [24]. FOS also plays an essential role in the process of myogenic differentiation. FOS inhibits mice myoblast myogenesis by binding to the putative cAMP response element on the MyoD promoter and inhibiting MyoD expression in dividing myoblasts [12]. Conversely, MyoD binding site overlaps with the serum-responsive element in the FOS promoter, and then MyoD acts as a negative regulator for FOS transcription by blocking serum responsiveness through this binding site [13]. In this study, we found that miR-501-3p promotes myoblast differentiation by inhibiting FOS expression. 

MDFI is a myogenic regulatory factor. MDFI blocks the nuclear localization signal of MyoD family protein resulting in the retention of transcription factors in the cytoplasm. MDFI inhibits mouse embryonic fibroblast cell line NIH-3T3 myogenic differentiated [25]. However, in our laboratory MDFI knock-in C2C12 model, we found that MDFI increased the expression of MyoD and promoted the differentiation of C2C12 cells through interaction with MyoD [14]. In addition, we found there is an FOS binding site in the MDFI promoter region through the database. In this study, we found that FOS banded to the MDFI promoter region by ChIP assay and that FOS inhibits MDFI expression by the dual-luciferase reporter gene. These results indicate FOS represses MDFI expression and the low level of MDFI inhibits C2C12 myogenic differentiation through MyoD.

In conclusion, our results show that miR-501-3p inhibited the C2C12 cell proliferation and promoted the C2C12 cell differentiation by inhibiting the expression of its target gene FOS. Our results also discover, besides the MyoD and FOS feedback loop, miR-501-3p formed a regulatory loop with MyoD, FOS, and MDFI to regulate myoblast development. These results not only expand our understanding of the muscle myogenic development mechanism in which miRNA and genes participate in controlling skeletal muscle development, but also provide potential treatment strategies for skeletal muscle or metabolic-related diseases in the future.

## 4. Materials and Methods

### 4.1. C2C12 Transfection and Differentiation

C2C12 were seeded in a cell plate at the density of 3 × 10^4^ cells/cm^2^. C2C12 were transfected with pCDNA3.1-FOS (Appendix A), siRNA-FOS (5′-AAAATAAACTCCAGTTTTTCCTT-3′), miR-501-3p inhibitor, miR-501-3p mimics, or control by Lipofectamine 3000 (Invitrogen, Carlsbad, CA, USA), according to the manufacturer’s instructions. We used qRT-PCR to detect transfection efficiency after each transfection. We transfected the C2C12 when the cells reached approximately 60% confluency. The medium was replaced with new growth medium 6 h later, and cells were maintained in the growth medium for an additional 24 h before myogenic differentiation induction. When cells reached 90% confluence, the cells were maintained in the differentiation medium (2% house serum) for seven days differentiation. MiR-501-3p mimics and inhibitor were purchased from GENEWIZ (Suzhou, Jiangsu, China), and the control was the scramble sequence negative control provided by GENEWIZ. 

### 4.2. RNA Extraction and PCR Analysis

Methods used for the RNA extraction and PCR analysis have been described previously [26]. The relative expression of mRNAs and microRNA was normalized with β-actin or U6 levels using the 2^−ΔΔ*C*t^ method [27]. U6 is a widely used normalizer for miRNA studies [28,29]. 2^−ΔΔ*C*t^ is defined as the ratio of the relative mRNA or miRNA level between the experimental group and the control group. Primers were designed using Primer Premier 5 according to the mice genes sequence obtained from NCBI. All the primers used in this study are shown in Table 1.

### 4.3. Edu Labeling

The transfected C2C12 were incubated in the growth medium for 24 h and then were used for Edu labeling by Cell-Light™ Edu Apollo^®^ 488 In Vitro Imaging Kit (RiboBio, Guangzhou, Guangdong, China) according to the manufacturer’s instructions. The Edu-labeled C2C12 were observed and recorded using a Nikon TE2000-U inverted microscope (Nikon Instruments, Tokyo, Japan). The Edu-positive C2C12 were counted using Image Pro Plus (Media Cybernetics, Rockville, MD, USA).

### 4.4. Immunofluorescence

After the transfected C2C12 myogenic differentiation induction for seven days, the cells were incubated in 4% paraformaldehyde for 10 min, 0.5% Triton X-100 for 5 min, and 5% FBS for 1 h at room temperature. Then, the cells were incubated with Myosin antibody at 4 °C overnight followed by incubation with Cy3-labeled secondary antibody at room temperature for 1 h. The fluorescence was observed using Nikon Eclipse Ti-s microscopy (Nikon, Tokyo, Japan). The cell nuclei were stained for DAPI (Beyotime, Nanjing, Jiangsu, China). More than six fields of view were captured in each cell well.

### 4.5. Western Blot Analysis

The method used for the Western blot analysis has been described previously [26]. The protein quantification was done using Image J v1.52 (NIH, Bethesda, MA, USA). The antibodies used in this study are listed in Table 2.

### 4.6. Bioinformatics Analysis 

The target gene prediction was conducted using the software mirTargets 1.2 in conjunction with TargetScan, MicroCosm, Pictar, and miRDB databases. The prediction of transcription factor binding site in the gene promoter region was through JASPAR databases.

### 4.7. Luciferase Reporter Assay

FOS 3′ UTR sequence was amplified and inserted into the pmirGLO Vector (Ambion, Carlsbad, CA, USA). For the luciferase reporter assay, HEK 293T cells were cotransfected with pmirGLO-FOS-3′ UTR plus either miR-501-3p mimics or control for 48 h. Either pmirGLO or pmirGLO-FOS-3′UTR-mut was used as a control for pmirGLO-FOS-3′UTR. The activities of firefly and Renilla luciferases were determined using the Dual-Luciferase Reporter Assay System (Promega, Madison, WI, USA), and firefly luciferase values were normalized to that of Renilla luciferase.

A ~1500 bp sequence upstream of MDFI gene in genomic DNA was amplified and inserted into pGL3-basic Vector (Ambion, Carlsbad, CA, USA). For the luciferase reporter assay, HEK 293T cells were cotransfected with pGL3-basic-MDFI upstream, pRL-TK, plus either pCDNA3.1-MyoD or pCDNA3.1. Either pGL3-basic-MDFI upstream-mut or pGL3-control was used as a control for pGL3-basic-MDFI upstream. The activities of firefly and Renilla luciferases were determined using the Dual-Luciferase Reporter Assay System (Promega, Madison, WI, USA), and firefly luciferase values were normalized to that of Renilla luciferase.

A ~1500 bp sequence upstream of miR-501-3p host gene Clcn5 in genomic DNA was amplified and inserted into pGL3-basic Vector (Ambion, Carlsbad, CA, USA). For the luciferase reporter assay, HEK 293T cells were cotransfected with pGL3-basic-Clcn5 upstream, pRL-TK, plus either pCDNA3.1-MyoD or pCDNA3.1. Either pGL3-basic-Clcn5 upstream-mut or pGL3-control was used as a control for pGL3-basic-Clcn5 upstream The activities of firefly and Renilla luciferases were determined using the Dual-Luciferase Reporter Assay System (Promega, Madison, WI, USA), and firefly luciferase values were normalized to that of Renilla luciferase.

### 4.8. ChIP Assay

The ChIP-IT^®^ Express Magnetic ChIP Kit & Sonication Shearing Kit (Catalog number 53008) was purchased from Active Motif (Carlsbad, CA, USA). For ChIP assay, the DNA was immunoprecipitated with the MyoD or FOS antibody, and ChIP analysis was performed according to the manufacturer’s protocol. DNA samples before immunoprecipitation were used as a template for input control. Primers used for ChIP assay are shown in Table 1. The antibodies used for ChIP assay are shown in Table 2.

### 4.9. Statistical Analysis

All data are expressed as the mean ± standard error of the mean (S.E.M.), and at least three replicates were used per group. The SPSS analysis results show that our data is a normal distribution, and homogeneity of data between each treatment group is equal. Significant differences between treatment groups were determined by one-way ANOVA (SPSS v18.0, IBM Knowledge Center, Chicago, IL, USA). Significance was achieved when *p* < 0.05. * is *p* < 0.05, **** is *p* < 0.01, and *** is *p* < 0.001.

## Figures and Tables

**Figure 1 cells-08-00573-f001:**
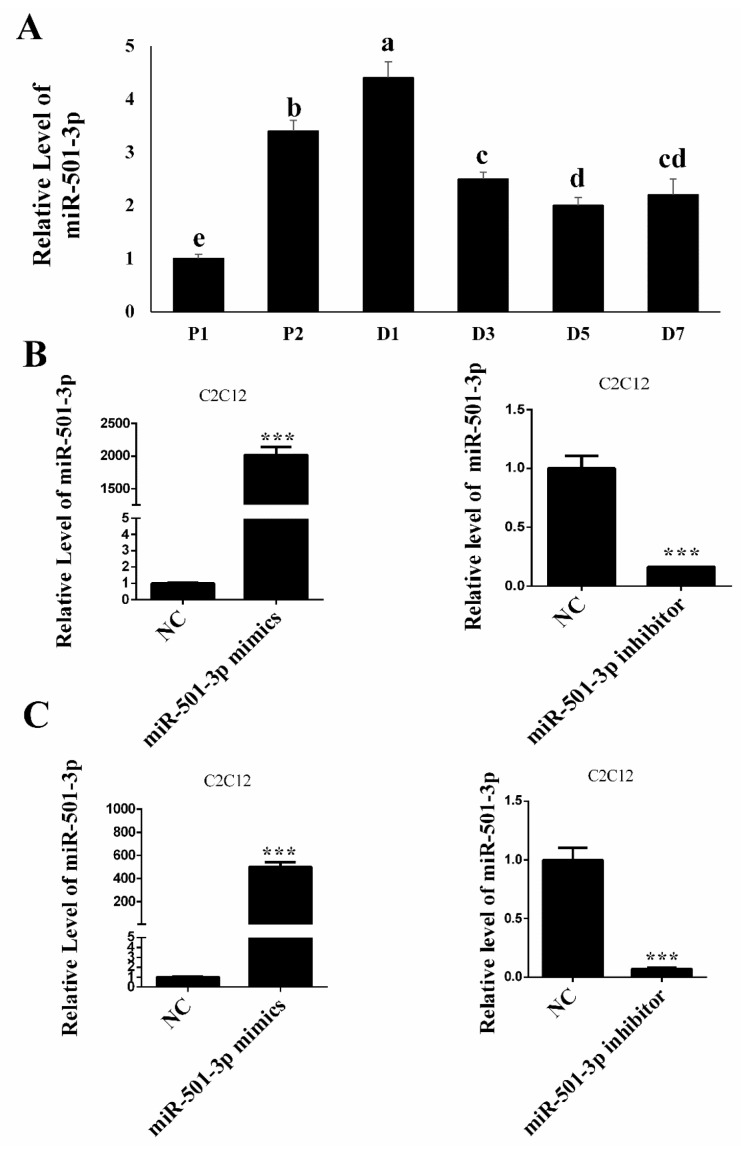
The miR-501-3p level in C2C12. (**A**) miR-501-3p levels at different stages of C2C12 myogenic development. (**B**) On the C2C12 proliferation day 1, miR-501-3p was elevated or inhibited by the miR-501-3p mimics or inhibitor, respectively. (**C**) On differentiation day 7, miR-501-3p was elevated or inhibited by the miR-501-3p mimics or inhibitor, respectively. NC is the negative control for the miR-501-3p mimics or inhibitor. P1: proliferation day 1, D1: differentiation day 1. In Figure 1A, bars with different letters indicate they are statistically different (*p* < 0.05). *** *p* < 0.001. The results are presented as mean ± S.E.M. of three replicates for each group.

**Figure 2 cells-08-00573-f002:**
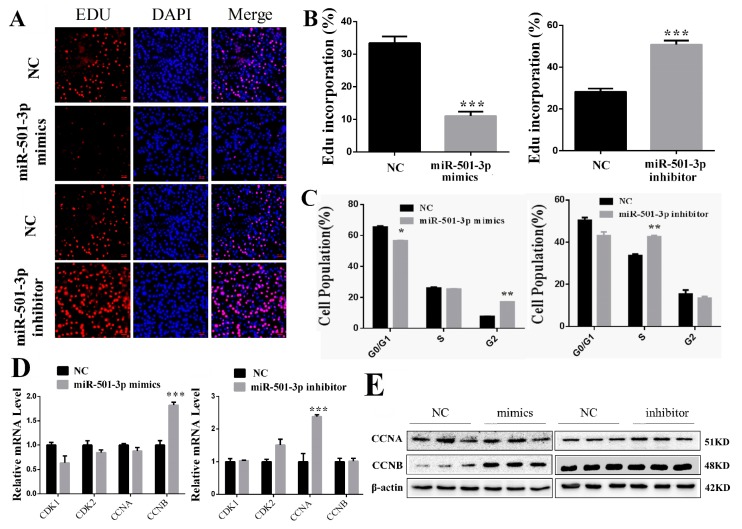
MiR-501-3p inhibits C2C12 proliferation. (**A**). MiR-501-3p mimics lowered the percentage of Edu-positive C2C12, and miR-501-3p inhibitor increased the percentage of Edu-positive C2C12. Representative images of the immunofluorescent staining for proliferating C2C12 are shown. Proliferating C2C12 were labeled with Edu fluorescent dye (red). (**B**) The quantitative data of proliferating C2C12 in Figure 2A. (**C**) The different cell phases analyzed by flow cytometry. (**D**) qRT-PCR confirmed miR-501-3p is negatively correlated with proliferation-related genes in C2C12 transfected with miR-501-3p mimics and inhibitor. (**E**) Western blot result shows that the protein level corresponded to the mRNA result. * *p* < 0.05, ** *p* < 0.01, *** *p* < 0.001. The results are presented as mean ± S.E.M. of three replicates for each group. Magnification 100×. The scale bar on the photomicrographs represents 50 μm.

**Figure 3 cells-08-00573-f003:**
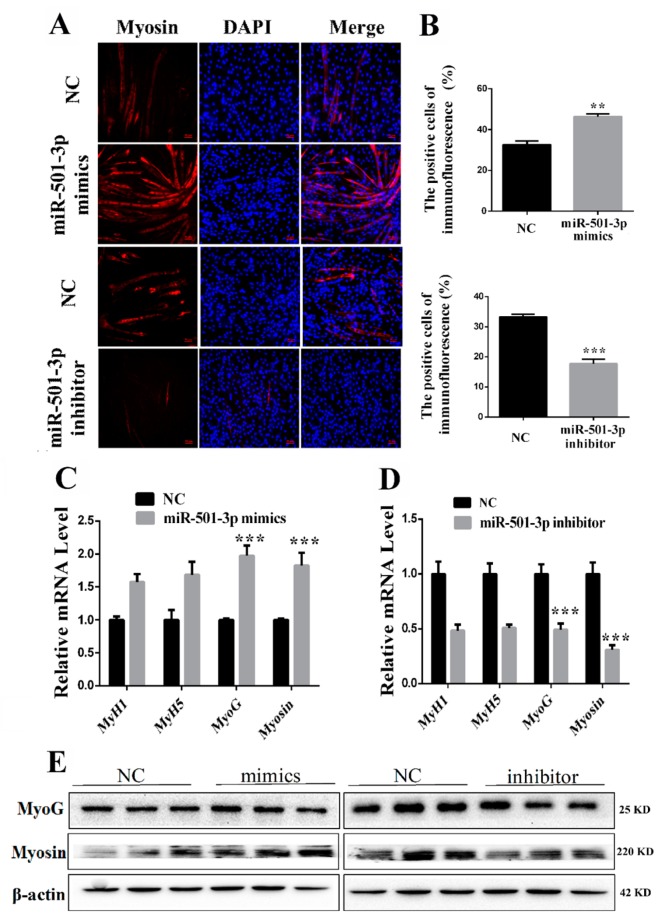
MiR-501-3p promotes C2C12 differentiation. We measured the expression of myogenic marker genes after C2C12 were transfected with miR-501-3p mimics or inhibitor. (**A**) MiR-501-3p mimics increased the percentage of Myosin-positive C2C12, and miR-501-3p inhibitor decreased the percentage of Myosin-positive C2C12. Representative images of the immunofluorescent staining for differentiated C2C12 are shown. Myosin: red, a molecular marker of myogenesis; DAPI: blue, cell nuclei; Merge: C2C12 fused into primary myotubes are shown in the insets. (**B**) The quantitative data of Myosin-positive C2C12 in Figure 3A. (**C**) The qRT-PCR confirmed miR-501-3p mimics increased (*p* < 0.05) mRNA level of myogenic marker genes after differentiation induction. (**D**) The qRT-PCR confirmed miR-501-3p inhibitor decreased (*p* < 0.05) mRNA level of myogenic marker genes after differentiation induction. (**E**) Western blot was used to measure the myogenic marker protein level after differentiation induction. ** *p* < 0.01; *** *p* < 0.001. The results are presented as mean ± S.E.M. of three replicates for each group. Magnification 100×. The scale bar on the photomicrographs represents 50 μm.

**Figure 4 cells-08-00573-f004:**
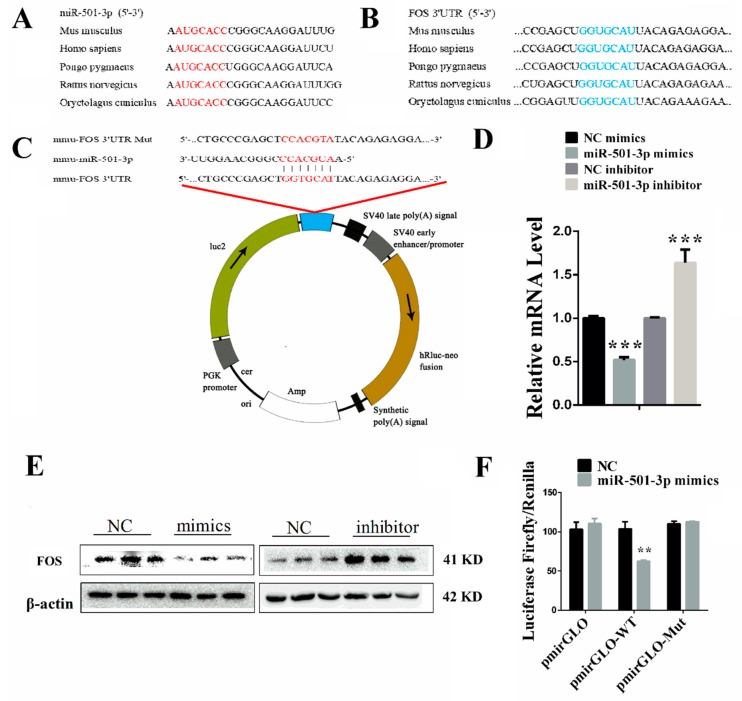
FOS is a direct target gene of miR-501-3p. (**A**) The sequence of mature miR-501-3p of different species. (**B**) The alignment result of FOS 3′UTR sequence from different species. Red: the complete complementary sequence of the FOS 3′UTR. (**C**) The complementary pairing of mmu-miR-501-3p with the targeted gene FOS 3′UTR or mutated UTR. (**D**) FOS mRNA level was negatively correlated with miR-501-3p level. (**E**) FOS protein level was negatively correlated with miR-501-3p level. (**F**) miR-501-3p inhibited the pmirGLO-*FOS* 3′ UTR vector luciferase activity. Luciferase reporters were transfected into HEK-293T cells with either miR-501-3p mimics or control. Luciferase activity was assayed 24 h after transfection. ** *p* < 0.01. *** *p* < 0.001. The results are presented as mean ± S.E.M. of three replicates for each group.

**Figure 5 cells-08-00573-f005:**
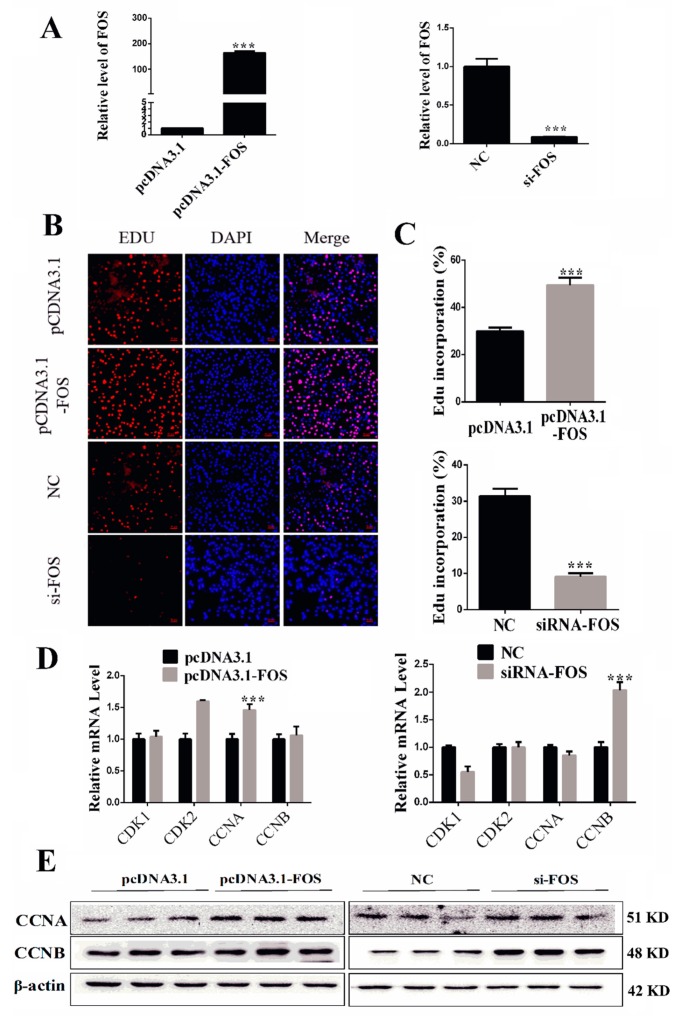
Overexpressing FOS promotes C2C12 proliferation. (**A**) Overexpressing or inhibiting FOS expression through transfecting C2C12 with pcDNA3.1(+)-FOS recombinant vector and siRNA-FOS, respectively. (**B**) Overexpressing FOS increased the percentage of Edu-positive C2C12, and interfering FOS decreased the percentage of Edu-positive C2C12. Representative images of the immunofluorescent proliferating C2C12 are shown. Proliferating C2C12 were labeled with Edu fluorescent dye (red). (**C**) The quantitative data of proliferating C2C12 number in Figure 5B. (**D**) qRT-PCR confirmed the mRNA level of proliferation-related genes in C2C12 transfected with pcDNA3.1(+)-FOS recombinant vector and siRNA-FOS, respectively. (**E**) Western blot result shows that the protein level corresponded to the mRNA result. *** *p* < 0.001. All the results are presented as mean ± S.E.M. of three replicates for each group. Magnification 100×. The scale bars on the photomicrographs represent 50 μm.

**Figure 6 cells-08-00573-f006:**
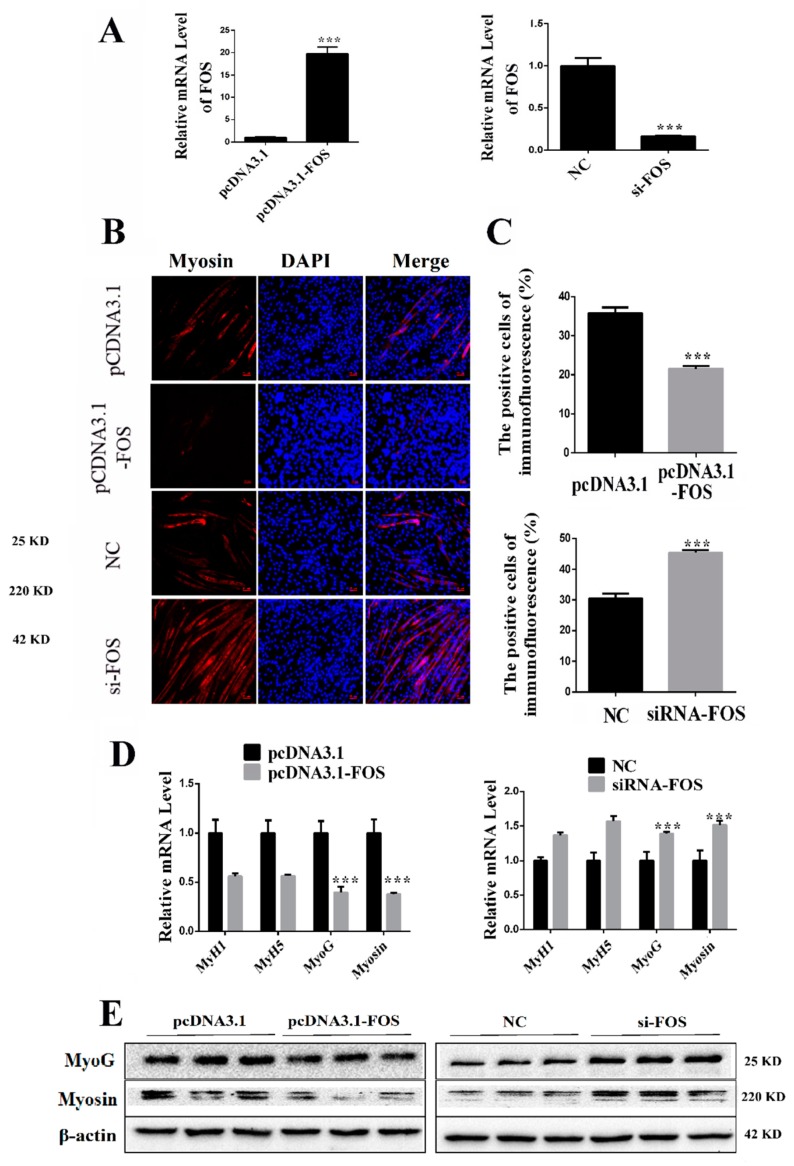
Overexpressing FOS inhibits C2C12 differentiation. (**A**) Overexpressing or inhibiting FOS expression through transfecting C2C12 with pcDNA3.1(+)-FOS recombinant vector and siRNA-FOS, respectively. (**B**) Overexpressing FOS decreased the percentage of Myosin-positive C2C12, and interfering FOS increased the percentage of Edu-positive C2C12. Representative images of the immunofluorescent staining for differentiated C2C12 are shown. Myosin: red, a molecular marker of myogenesis; DAPI: blue, cell nuclei; Merge: the C2C12 fused into primary myotubes are shown in the insets. (**C**) The quantitative data of Myosin-positive C2C12 in Figure 6B. (**D**) qRT-PCR confirmed the mRNA level of myogenic marker genes in C2C12 transfected with pcDNA3.1(+)-FOS recombinant vector and siRNA-FOS, respectively. (**E**) We used Western blot to measure the protein level of myogenic marker genes after differentiation induction. Western blot result shows that the protein level corresponded to the mRNA result after differentiation induction. *** *p* < 0.01. The results are presented as mean ± S.E.M. of three replicates for each group. Magnification 100×. The scale bars on the photomicrographs represent 50 μm.

**Figure 7 cells-08-00573-f007:**
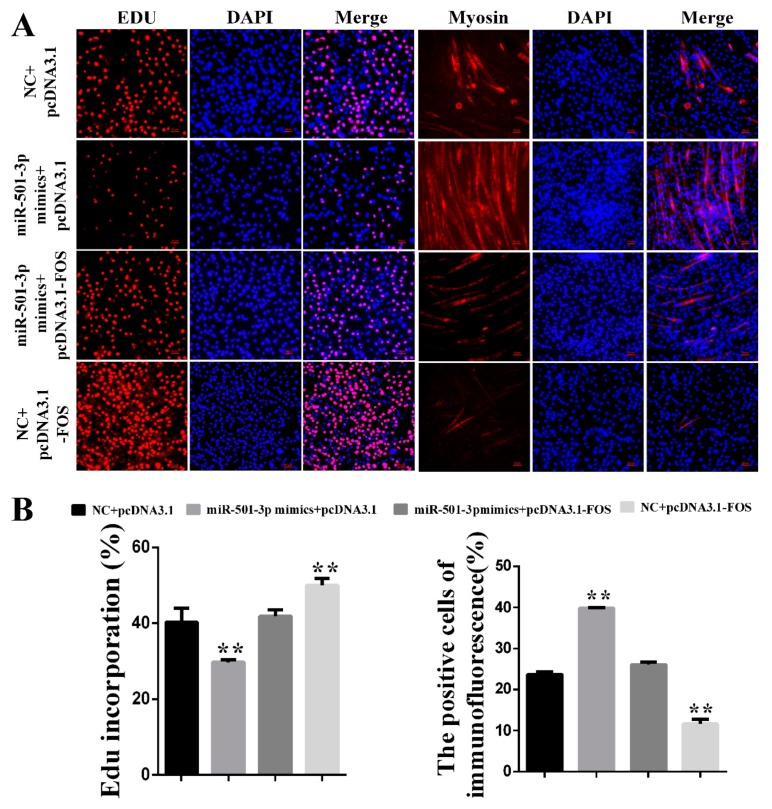
MiR-501-3p promotes C2C12 myogenesis by targeting FOS. (**A**) MiR-501-3p function on C2C12 proliferation and differentiation requires FOS. Representative images of the immunofluorescent staining for proliferating C2C12 and differentiated C2C12 are shown. (**B**) The quantitative data of proliferating and differentiated C2C12 number in Figure 7A. ** *p* < 0.01. The results are presented as mean ± S.E.M. of three replicates for each group. Magnification 100×. The scale bars on the photomicrographs represent 50 μm.

**Figure 8 cells-08-00573-f008:**
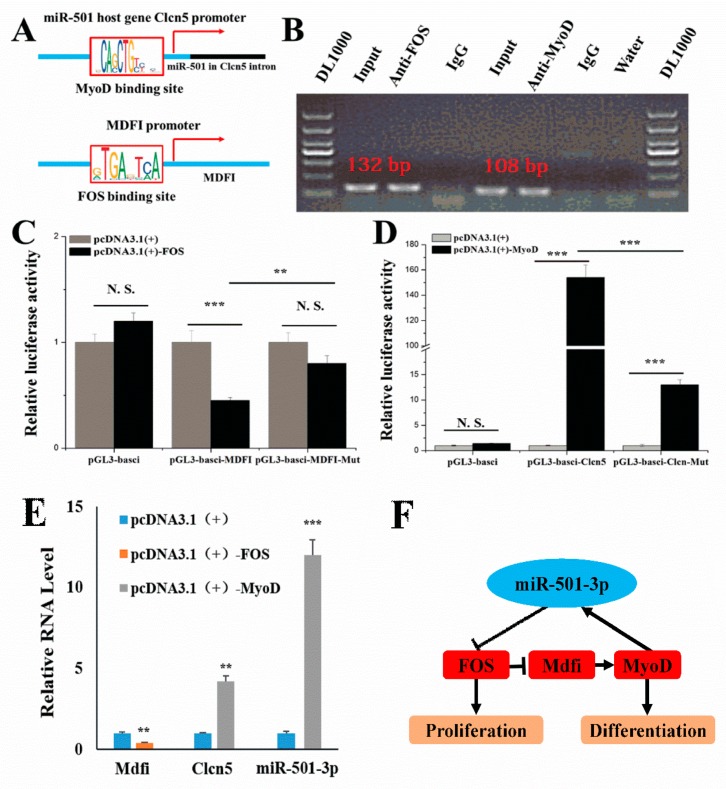
**miR-501-3p formed a feedback loop with FOS, MDFI, and MyoD**. (**A**) Schematic representation of MyoD binds at the upstream of miR-501-3p in the genome, and FOS binds at the upstream of MDFI in the genome. (**B**) ChIP experiments were used to verify that FOS and MyoD bound to the promoter regions of MDFI and Clcn5, respectively. (**C**) Luciferase reporters show FOS bound to the MDFI promoter region. (**D**) Luciferase reporters show MyoD bound to the Clcn5 promoter region. (**E**) QPCR results indicate that FOS inhibited the expression of MDFI and MyoD increased the expression of Clcn5 and miR-501-3p. (**F**) A feedback loop formed by miR-501-3p, FOS, MDFI, and MyoD to regulate myogenic differentiation. **, *p* < 0.01; ***, *p* < 0.001. The results are presented as mean ± S.E.M. of three replicates for each group.

**Table 1 cells-08-00573-t001:** All the primers used in this study.

Gene Name	Forward Primer Sequence (5′-3′)	Reversed Primer Sequence (5′-3′)
U6	CTCGCTTCGGCAGCACA	AACGCTTCACGAATTTGCGT
β-actin	GGCTGTATTCCCCTCCATCG	CCAGTTGGTAACAATGCCATGT
FOS	GACAGCCTTTCCTACTACCAT	CTTATTCCGTTCCCTTCG
FOS 3′ UTR	TCGACAGACGTGCCACTGCCCG	CTAGGGGAAGACGTGTTTCTCC
MyoD ChIP	GACTACCCTATTGCGGTTGCG	GCCACGGAGGGCCCGCT
FOS ChIP	TTTTGAAGGCAGGTGAGGC	GAGTTTGGGAGCACTCCTC
CDK2	GTGGTACCGAGCACCTGAAA	CGGGTCACCATTTCAGCAAA
CDK1	ACAGAGAGGGTCCGTCGTAA	ATTGCAGTACTGGGCACTCC
CCNA	CATCTCACTACATAGCTGACTT	GTGGCGCCTTTAATCCCAGA
CCNB	GAAACGCATTCTCTGCGACC	ATTTTCGAGTTCCTGGTGACT
MyH1	CATCCCTAAAGGCAGGCTCT	GAGCCTCGATTCGCTCCTTT
MyH5	GCCAGCATGGAGACGATACA	TCGGTCTCGTACTTGGTCCT
MyoG	GAGACATGAGTGCCCTGACC	AGGCTTTGGAACCGGATAGC
Myosin	GCTGAAGAAGAGCAGTTTCCG	TCACACTCAAACTCCACCCG

**Table 2 cells-08-00573-t002:** The antibodies used in this study.

Primary Antibody	Clone	Company	Catalog No.	Dilution
FOS	Monoclonal	CST	ab54481	1:2000
Myosin	Polyclonal	Bioss	bs-10906R	1:2000
MyoG	Polyclonal	Santa Cruz	sc-71629	1:500
CCNA	Monoclonal	CST	4656T	1:1000
CCNB	Polyclonal	CST	4138T	1:1000
MyoD	Polyclonal	Santa Cruz	sc-71629	1:500
β-Actin	Monoclonal	Bioss	bsm-33036M	1:1000
Secondary antibody	Conjugate Used	Company	Catalog No.	Dilution
Goat Anti-rabbit IgG	HRP	Bioss	bs-0295G	1:10,000
Goat Anti-mouse IgG	HRP	Bioss	bs-40296G	1:10,000

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
