# Peer review of "MiR-501-3p Forms a Feedback Loop with FOS, MDFI, and MyoD to Regulate C2C12 Myogenesis"

_cells, 2019, doi:10.3390/cells8060573_

Round 1
Reviewer 1 Report
This manuscript tries to examine the function of miR-501-3p in the proliferation and differentiation period of C2C12 cells by transfecting cells with miR-501-3p mimic and inhibitor. Then the authors confirmed there is a direct regulatory relationship between MyoD and miR-501-3p, FOS and MDFI through QPCR, dual luciferase reporter system, and ChIP experiments.
All experiments in this manuscript are properly designed and the results support the conclusion. This manuscript will expand our understanding of miRNAs and genes work collaboratively in regulating skeletal muscle development.
There are some minor comments to be considered.
Some parts of the figure legends for Fig. 1, 2, 3 does not correspond to the contents of the figure. For example, there is no data for flow cytometry in Fig. 2 although there is a description “C. The different cell phase analyzed by flow cytometry” in the legend for Figure 2.
Culture conditions for differentiating C2C12 cells should be described in detail.
Need to provide information about the sequence of the siRNAs.
Information on pcDNA3.1-FOS should be provided. Figure 7 shows that ectopic FOS expression was suppressed by miR-501-3p mimics. Considering that miR-501-3p binds to the 3'-UTR of FOS (Figure 4), does pcDNA3.1-FOS also contain the 3'-UTR?
Author Response
Point 1: Some parts of the figure legends for Fig. 1, 2, 3 does not correspond to the contents of the figure. For example, there is no data for flow cytometry in Fig. 2 although there is a description “C. The different cell phase analysed by flow cytometry” in the legend for Figure 2.
Response 1: We apologize for our carelessness in preparing the original manuscript. We have made the corrections in the revised manuscript (lines 78-79 of page 2, line 81 of page 2, line 86 of page 3, line 106 of page 4, lines 137-139 of page 6).
Point 2: Culture conditions for differentiating C2C12 cells should be described in detail.
Response 2: We have added the C2C12 differentiation culture conditions in the revised manuscript (line 320 of page 14, lines 326-328 of page 15).
Point 3: Need to provide information about the sequence of the siRNAs.
Response 3: We have added the FOS siRNA sequence to materials and methods section in the revised manuscript (line 321 of page 14).
Point 4: Information on pcDNA3.1-FOS should be provided. Figure 7 shows that ectopic FOS expression was suppressed by miR-501-3p mimics. Considering that miR-501-3p binds to the 3'-UTR of FOS (Figure 4), does pcDNA3.1-FOS also contain the 3'-UTR?
Response 4: We have added the pcDNA3.1-FOS information to the materials and methods section in the revised manuscript (line 321 of page 14). We inserted only the FOS coding sequence to pcDNA3.1. The pcDNA3.1-FOS 3'-UTR is a BGH polyA sequence. Bioinformatics analysis indicated there is no miR-501-3p binding site in the BGH polyA sequence.
Reviewer 2 Report
The paper written by Lianjie Hou et al. shows that miR-501-3p inhibits C2C12 cell proliferation targeting FOS. Moreover, the authors propose that miR-501-3p forms a regulatory loop with MyoD, Fos, and MDFI to regulate myoblast development. The study is well designed and executed. It provides convincing data, and the statistical analysis is well performed. Novel and original findings that could be of great interest for the international scientific community are reported.
Minor point: please check the results (section 2.1) and the legend to figure 1: sentences about miR-22 are not clear to the reader. Are they right?
please check the results (section 2.8) and the legend to figure 8: "MDFI and MyoD inhibited the expression of Clcn5 and miR-501-3p", is it in accordance with the showed results?
Author Response
Point 1: Please check the results (section 2.1) and the legend to figure 1: sentences about miR-22 are not clear to the reader. Are they right?
Response 1: We apologize for our carelessness in preparing the original manuscript. We have made the corrections in the revised manuscript (lines 78-79 of page 2 and line 86 of page 3).
Point 2: Please check the results (section 2.8) and the legend to figure 8: "MDFI and MyoD inhibited the expression of Clcn5 and miR-501-3p", is it in accordance with the showed results?
Response 2: We apologize for our carelessness in preparing the original manuscript. We have made the corrections in the revised manuscript (line 255 of page 12 and line 264 of page 13).